# The Influence of Gender and the Specificity of Sports Activities on the Performance of Body Balance for Students of the Faculty of Physical Education and Sports

**DOI:** 10.3390/ijerph19137672

**Published:** 2022-06-23

**Authors:** George Danut Mocanu, Gabriel Murariu, Ilie Onu, Georgian Badicu

**Affiliations:** 1Faculty of Physical Education and Sport, “Dunarea de Jos” University of Galati, 47 Domneasca Street, 800008 Galati, Romania; george.mocanu@ugal.ro; 2Faculty of Sciences and Environment, “Dunarea de Jos” University of Galati, 47 Domneasca Street, 800008 Galati, Romania; 3Department of Biomedical Sciences, Faculty of Medical Bioengineering, University of Medicine and Pharmacy “Grigore T. Popa” Iasi, 700454 Iasi, Romania; ilie.onu@umfiasi.ro; 4Department of Physical Education and Special Motricity, Transilvania University of Brasov, 29 Eroilor Blvd., 500036 Brasov, Romania

**Keywords:** sports activity, university students, static and dynamic balance, evaluation, gender differences

## Abstract

A sense of balance is required in sports activities, conditioning the quality of movements and physical performance. (1) *The purpose of the study* is to investigate the influence of gender and the specificity of sports activities on body balance. The investigated *participants* are 157 students of the Faculty of Physical Education and Sports/Bachelor’s degree: 109 men (age = 20.49 ± 2.03, body mass index, BMI = 22.96 ± 3.20), and 48 women (age = 20.21 ± 1.51, BMI = 21.05 ± 2.78). (2) Design: Cross-sectional study, with the definition of the variables gender and sport activity with three stages (non-athletes/NA, team sports games/TSG, and individual sports/IS). The *evaluation* was based on four dynamic balance tests (Bass test/points, Functional reach test/cm, Fukuda test/degrees of rotation, and Walk and turn field sobriety test/errors) and three static balance tests (Flamingo test/falls, Stork test, and One-leg standing test with eyes closed/s). (3) Results: The variance analysis (multivariate and univariate tests) indicates the superiority of women in most tests applied, but with significantly better values (*p* < 0.05) only for the Flamingo test and Bass test. Men have superior results only for vestibular stability (Fukuda test) and One-leg standing test, but it is statistically insignificant (*p* > 0.05). The TSG group has slightly better values than the IS group for the whole set of tests conducted, but these are not statistically significant (*p* > 0.05), so we cannot highlight the certain superiority of TSG practitioners over those involved in IS. Both the TSG and the IS group outperformed all tests compared to the NA group, with significant differences (*p* < 0.05), especially for the TSG. *Conclusion*: Women have better values than men on most tests, and performance sports students have higher average scores than those in the NA group, which demonstrates the beneficial influence of specific training on static and dynamic postural stability.

## 1. Introduction

### 1.1. Factors Influencing the Manifestation of Balance

Balance is an ability that brings together somatic-sensory, vestibular and visual information for superior postural control, which is extremely required in sports activities and which conditions motor performance [1]. In maintaining balance, young dancers use mainly the somatic-sensory system, while experienced adults use the vestibular system without reporting any differences between women and men [2]. Kinesthetic functions at the level of the neck have a role in maintaining balance and oculomotor control for elite hockey players, according to [3]. Cervical proprioception supplement information is provided by the vestibular and auditory analyzers as an adaptation for Taekwondo fighters [4]. Maintaining dynamic balance performances are significantly improved by the presence of visual feedback [5]. Poor ballet dancers in Poland show poor performance of static and dynamic balance with their eyes closed [6]. For young women in Poland (18–28 years old), the use of shoes with high heel/heel height (especially 10 cm), combined with the exclusion of visual function, generates a limitation of the values of static balance [7].

The influence of anthropometric variables on body stability generates some contradictory aspects in specialized research. Weak positive and negative correlations, or lack of correlations between anthropometric dimensions and the value of results in static and dynamic equilibrium tests, have been identified for women university students in Iran (18–25 years), so there is no single anthropometric indicator to justify variations in balance [8]. Another study identifies strong correlations between the value of static balance and the level of trunk muscle endurance for male university students in Tehran [9]. Even if the balance of the body is influenced by the strength of the muscles of the lower limbs, there are studies that indicate that toe flexor strength does not have a significant influence while maintaining static upright standing [10].

Physical effort affects the balance values for shooting athletes, with higher oscillations and speeds of the CoP (center of pressure) being found for the groups that made an effort, compared to the values of a group that was not physically requested before shooting, according to [11]. Muscle fatigue is a factor that influences the quality of postural control. Adolescent sports girls have a dynamic postural control superior to non-sports girls (medial, posteromedial, and posterior directions) after performing an effort for 20 min; this demonstrates that sports training has a beneficial role in adapting the body and maintaining body stability in conditions of fatigue [12]. Fatigue generates postural instability for both legs (dominant and non-dominant) in those who practice symmetrical and asymmetrical sports, but the disturbance of balance is greater for the dominant leg in those who are involved in asymmetrical sports [13]. Exercises of maximum anaerobic intensity generate severe fatigue, which also affects the values of dynamic balance. For a group of university athletes (23.75 years old), the installation of fatigue has negative effects on the quality of movements in all directions: anterior, posteromedial, and posterolateral, according to [14]. Skiing has been shown to affect the balance with fatigue, which poses an additional risk of injury, along with the use of skis with a large waist-width, especially on icy surfaces, according to [15].

Sedentary behavior is associated with impaired muscular strength, cardiorespiratory fitness, and balance for adults (>18 years), according to [16]. Another factor that influences postural control and walking efficiency is sleep quality [17]. Furthermore, the existence of groin pain for Tunisian football players generates an impairment of the results in the tests for assessing static and dynamic stability compared to a group in which these problems are not present [18].

### 1.2. Gender Differences in the Manifestation of Balance

The systematic review of the specialized studies did not firmly formulate the superiority of men or women in the tests of balance assessment; there are studies that confirm the better values for a gender but also research that does not find significant differences between genders [19]. A better balance in girls in static assessments is found for Polish university students specializing in Physical Education. After sleep deprivation for 24 h, men with higher oscillations of the CoP were more sensitive to lack of sleep [20]. Testing young people on the oscillating platform showed weaker stability of women, but only in the frontal (coronal) plane, not in the other planes, with men having a better average holding time on the platform, according to [21]. The posturographic analysis indicates differences in postural control between genders; women have lower stability, with higher oscillations and speeds of CoP, requiring superior neuromuscular efforts to maintain balance, according to [22]. Muscle fatigue decreases performance when assessing balance by the Star Excursion Balance Test (SEBT) for young adults, but higher values are found for women (greater contact distances and greater knee flexion), according to [23].

Female athletes have lower values of lower limb muscle strength but also compensatory models for neuromuscular control and higher values of static balance compared to male athletes [24]. Unicycling training for university students specializing in Physical Education and Sports in Croatia (5 weeks) is significantly effective in terms of dynamic balance for both genders, but significant improvements in static balance are only noticeable for women [25]. The use of Eyerobics Visual Skills Training Program, for 4 weeks, on students of the Faculty of Physical Education and Sports (19–23 years old) has generated the improvement of static balance, and women have higher values than men [26].

Postural balance in young women is similar to that of young men, but older men have higher body balance and greater balance problems compared to women in the same age group [27].

### 1.3. The Relationship between Physical Activity and Balance

For participants experienced in sports activities with asymmetrical demands, the beneficial influence of a warm-up on the balance values is noticeable only for the dominant leg. However, performing the 10-min warm-up on the cycle ergometer does not reduce the differences/asymmetries related to the balance between the dominant and the non-dominant leg in healthy young athletes [28]. The significant correlation between superior dynamic postural stability and the values of flexibility in young people (20–29) indicates their importance in athletic performance and ensuring bodily integrity in daily activities [29]. The interview with the Olympic shooters identified the fact that fitness training (strength and endurance for the arms and abdominal muscles) is important in optimizing performance, but general balance training is neglected [30].

The time interval of isolation and physical inactivity during the COVID-19 pandemic has led to a decrease in balance performance for children in Spain, but without significant differences between team sports, individual, and physically inactive, according to [31]. The comparison between young sportswomen and sedentary young women in Turkey showed significantly lower values of sedentary women in static and dynamic balance tests. For athletes, women basketball players have the best results in dynamic balance, and for the static one (the Flamingo test), the top are the female volleyball and football players [32]. Higher balance values for college male students in Croatia (x = 21.6 years) involved in sports are a good predictor of performance in agility tests, so balance training should be incorporated into the training of these subjects [33]. Female children practicing modern dance have higher values of static and dynamic balance compared to physically active similar groups that are involved in other sports activities [34].

Balance training exercises using different balance tools are more effective than resistance training programs in terms of ankle sprain improvement results. Higher values of dynamic balance, pain relief, and return to sports activities were noted for the subjects involved (20–39 years) [35]. The implementation of programs based on the stimulation of the proprioceptive sense in the training of female football payers has generated an improvement in the performance of agility and static balance tests, according to [36]. The use of strength training for male volleyball players in Turkey (18–25 years old) has a positive effect on the values of static and dynamic balance [37]. The application of WBV (whole-body vibration) programs, for 8 weeks, for students from physical education and sports faculties in Greece has generated favorable effects on static balance, especially the 60 s variant, according to [38].

For teenage rugby players in Poland, significant associations are determined between the values of dynamic balance (evaluated by Y-Balance Test) and the strength of the lower limbs at different jumps, these two interdependent variables having a role in reducing the risk of injury [39].

*The purpose of the study* is to identify how gender and the inclusion of students in different sports activities (team sports, individual sports, and subjects not involved in performance sports—but physically active) generate differences in performance on balance tests and reporting these values to similar research conducted nationally and internationally.

*Working hypotheses*:

**Hypothesis** **1** **(H1).***We assume that the independent variable gender will generate significant differences in the tests for the assessment of static and dynamic balance*.

**Hypothesis** **2** **(H2).***We estimate that we will obtain statistically significant differences between the average values of performance athletes (team sports games group vs. individual sports group)*.

**Hypothesis** **3** **(H3).***We estimate that the group of athletes practicing team sports games will have statistically significantly superior performance compared to the group of non-athlete students in the balance assessment tests*.

**Hypothesis** **4** **(H4).***We estimate that the group of individual sports practitioners will perform statistically significantly better compared to the group of non-athlete students for the entire set of tests used*.

## 2. Materials and Methods

### 2.1. Participants

The students involved in the study belong to the Faculty of Physical Education and Sports (with specializations in Physical Education and Sports, Physical Therapy, and Special Motor Skills) within the “Dunărea de Jos” University of Galați. All 164 subjects belonging to year 1 of the bachelor’s degree were invited to participate in the research, out of which 157 students remained in the study, with 7 cases being eliminated from the initial group (those who did not pass all the tests from the applied set of tests, those who were injured, or did not respond favorably to the invitation). The distribution of subjects by gender is 109 men (age = 20.49 ± 2.03, weight = 73.22 ± 11.83, waist = 178.33 ± 7.09, body mass index/BMI = 22.96 ± 3.20), and 48 women (age = 20.21 ± 1.51, weight = 56.75 ± 9.13, waist = 164.00 ± 7.32, BMI = 21.05 ± 2.78). The constitution of the groups according to the involvement and the type of sports activity was performed on 3 distinct categories: NA/non-athletes (43.95%—those who do not do performance sports but are physically active, involved in leisure physical activities, and in the practical curricular ones), the athletes involved in TSG/team sports games (soccer 15.92%, basketball 3.82%, handball 3.82%, volleyball 2.55%, rugby 1.27%), and IS/individual sports (fitness + bodybuilding variants 10.83%, athletics—running 4.46%, martial arts 3.82%, boxing 2.55%, sport dancing 2.55%, tennis and table tennis 1.91%, kayaking and rowing 1.27%, swimming 1.27%). The number of subjects for the independent variables listed, as well as the percentage values related to them, are summarized in Table 1. All the investigated students presented a favorable medical opinion for performing the physical effort.

### 2.2. Procedure

The applied research variant is one that falls into the category of cross-sectional studies. The measurement and testing of the investigated group were planned for April 2019, a period preceding the onset of the COVID-19 pandemic. The evaluation of the study group took place within the Research Center for Human Performance, associated with the Faculty of Physical Education and Sports in Galați. The investigation of the balance was performed by 7 tests, of which 4 are oriented for the evaluation of the dynamic balance (Bass test, Functional reach test, Fukuda test, and Walk and turn field sobriety test), and 3 for the evaluation of static balance (Flamingo test, Stork test, One-leg standing test with eyes closed), additional data related to the test methodology, interpretation of results, validity, and reliability of the set of tests being stipulated by [40,41,42,43,44,45,46,47,48]. A brief overview of the motor tasks associated with each test and how to quantify their results facilitates an understanding of the specific efforts to maintain balance:*One-leg standing balance test*: Standing on one leg, the other leg is raised with the knee bent, the forearms crossed on the chest (There is also the option of fixing the raised leg behind the knee of the lower support member/popliteal space). The time maintaining balance with eyes closed is timed (s).*Stork test:* Sitting on one leg with palms on hips and eyes open, raise their other leg and rest their toes on the opposite knee. Then move on to the tip of the foot remaining on the ground, and the time maintaining the balance is timed (s). The stopwatch is stopped when unbalanced, moving the supporting leg, contact with the whole sole, the lack of contact with the knee, the movement of the arms.*Flamingo test* (Eurofit test): The assessor tries to maintain balance on one leg (on a beam 50 cm long, 4 cm high, and 3 cm wide), the arm on the same side is raised, the other lower limb is flexed, raised back and caught by the hand on the same side. We record the number of attempts/failures required to accumulate 60 s of holding the position.*Functional reach test*: The performer is standing sideways against the wall, with the same arm outstretched forward and parallel to the ground, and the examiner draws a mark on the wall at the end of the middle finger. The performer then bends their torso forward to the point where the balance can be maintained and marks the place where the middle finger reaches. The difference between the two drawn marks is recorded in cm.*Bass test:* It involves performing 10 consecutive jumps from one foot to the other, landing on the forefoot and maintaining the position for 5 s, treading signs measuring 2.54 × 2 cm. For each correct landing, 5 points are awarded, plus 1 point for each second of holding, so the maximum score is 100.*Walk and Turn Field Sobriety test:* Evaluates the balance while walking along a straight line drawn on the ground. Take 9 steps with the heel of the foot placed in front, touching the toes of the back foot. After taking the steps, the subject must turn on one leg and return in the same way in the opposite direction. Note: The following errors are identified: “Can’t keep balance during instructions, Starts too soon, Stops walking, Misses heel-to-toe, Steps off line, Uses arms for balance, Improper turn, Incorrect number of steps.”*Fukuda test:* The subject moves on the spot/50 steps with the alternative lifting of the knees, with the arms outstretched forward and being blindfolded/with eyes closed. At the end, the angle of deviation from the initial direction (left or right) is measured; values > 30 degrees indicate problems of balance/labyrinthine damage on the side where the deviation occurred.

The evaluation process was planned between 12.00 and 16.00, as we wanted to avoid variations in performance values for the balance tests, especially their decrease if the tests were performed early in the morning or late in the evening [49,50,51]; as there are sources of fluctuations in performance for elite athletes generated by the circadian rhythm [52]. The investigation teams were reduced in number (6–8 people) in order to ensure a calm climate and to avoid disturbing factors with negative effects on performance due to the decrease in the ability to focus [53]. In order to avoid testing students during nervous or physical fatigue, they were advised to avoid engaging in vigorous physical activity prior to testing. Thorough training of the participants was carried out in connection with the role of the tests, the correct execution technique, and the ways of interpreting the scores obtained at each test, and they were allowed to familiarize themselves with the set of tests. In the warm-up part and in the breaks between tests, light physical efforts were made, based on low-intensity aerobic loads and stretching techniques, in order to ensure optimal muscle tone. The rules of deontology of scientific research, related to the protection and confidentiality of personal data for studies involving human subjects according to the Helsinki Declaration, were followed [54,55].

### 2.3. The Statistical Analysis of Data

The statistical processing of the data obtained after the evaluation was performed with the help of SPSS Software (Statistical Package for the Social Sciences—Vers.24). The ANOVA parametric techniques (multivariate and univariate tests) with factorial design (2 × 3) were used, defining the independent variables gender (with both sexes) ands sports activity with 3 stages (non-athletes, team sports games, and individual sports). Levene’s Test of Equality of Error Variances was used and the influence of each independent variable, the interaction between them on the whole set of tests and on each test separately, calculating the values of F and their significance thresholds, as well as the effect size expressed by Partial eta squared/Ƞ^2^_p_ was determined. Central trend indicators were calculated for each dependent variable (test), lower and upper limits of performance were identified, and differences in mean values were determined for the resulting data pairs (using Bonferroni Post Hoc Tests in interpreting the significance of the obtained differences) [56,57,58]. The confidence interval was set at 95% (*p* < 0.05) [59,60,61]. Graphs with average comparative values by genre for each class of sports activity (NA, TSG and IS) were generated using Microsoft Excel editor.

## 3. Results

The processed data are summarized in Table 2, Table 3, Table 4, Table 5 and Table 6 (multivariate/univariate test results and comparison of score pairs during the tests) in Figure 1, Figure 2 and Figure 3 (average values by gender for each step associated with the type of sports activity).

Multivariate analysis provides information on the overall effect of the two independent variables (gender and sports activity), but also on the interaction between them on all balance tests used (dependent variables) in Table 2. The gender variable and the sports activity variable generate statistically significant influences on the results of the equilibrium tests (*p* < 0.05), and with average values of size effect/Ƞ^2^_p_ (10.8% of the variance of the performances in the set of tests is explained by the gender variable, and 10.3% by the sports activity variable). However, the gender * sports activity interaction does not generate statistically significant results (*p* > 0.05), with a low value of Ƞ^2^_p_ (only 2.2% of the test performance variance is generated by this interaction).

For the independent variable gender, the univariate test analysis (Table 3) establishes its influence on the results in the case of each test within the applied evaluation set. It is observed that only in two cases statistically significant thresholds are reported (*p* < 0.05): the Flamingo test and Bass test, but with weak values of Partial Eta Squared (only 3% and 2.8% of the variance of the results in these tests is due to the independent variable gender). For the other five balance tests, only statistically insignificant influences of the gender variable are reported (*p* > 0.05), which are associated with very weak and even zero values of size effect/Partial Eta Squared.

Univariate test analysis at the level of the independent variable sports activity signals statistically significant influences (*p* < 0.05) for five of the seven balance assessment tests: One-leg standing balance test, Stork test, Flamingo test, Walk and turn field sobriety test, and Fukuda test. However, the size effect values expressed by Partial Eta Squared are moderate of even weak: 9.7% of the variance is due to the independent variable for the Fukuda test, 6.7% for the Stork test, 5.4% for the One-leg standing balance test, only 4.8% for the Walk and turn field sobriety test, and 3.9% for the Flamingo test. Statistically insignificant influences (*p* > 0.05) are reported only in the case of the Functional reach test and the Bass test, with low values of Partial Eta Squared.

The comparison of the average test results in pairs at the level of the independent variable gender (Table 5) indicates the superiority of women for five of the seven tests, but the existence of significant differences (*p* < 0.05) is confirmed only for the Flamingo test (where women need fewer attempts to accumulate the 60 s of balance on the support) and for the Bass test (where women also accumulate more points). The women’s group also has better scores on the Functional reach test (possibly also due to the superior flexibility of the gender at the spine level), the Stork test, and makes fewer mistakes in the Walk and turn field sobriety test, but without finding significant differences (*p* > 0.05). The group of men has superior average results for the One-leg standing balance test and shows better vestibular stability in the Fukuda test, but even in these two cases, the differences are insignificant (*p* > 0.05).

It should be noted the presence of eight cases with performances over 50 s for One-leg standing balance test in the case of men (with a group record of 89 s); for the group of women, there were only three cases with performances of over 20 s, an aspect that generated a lower average value for them in this test. Although for the Stork test, the top result of the whole lot (82.84 s) is also obtained by a man, the average value of men is slightly lower than that of women due to the fact that 32 men (29.35% of the whole lot) had results below 2 s, compared to the group of women, where only 5 cases (10.41%) had equally poor results. Significantly poorer male results for the Flamingo test result from 12 students (11%) needing more than 10 attempts to total the 60 s of balance on the support compared to the female group, where only 2 cases (4.16%) have similar results. This cancels out the slight advantage of men in terms of top results for this test: 16 cases (14.67%) manage to maintain the position in a single attempt, compared to 6 cases (12.5%) for the group of women. For the Bass test, women have higher percentage scores than men, 13 cases (27.08%) get values above 90 points, while 16 men (only 14.67%) fall into the same high values, but it should be noted that maximum performance (100 points) are, however, obtained only by 2 cases in the men’s group. For both sexes, we noticed the ease of detachment on one leg and precise landing with the coverage of chips placed on the ground, the problems being related to maintaining balance on the landing leg for 5 s, which limited the achievement of maximum scores for most participants.

For the Walk and turn field sobriety test, we found very good results in both genders, most of those tested ended the evaluation without identifying mistakes: 97 men (89%) and 44 women (91.67%). The rare errors found are related to small problems when turning the body or slightly crossing the line on which to travel. At the Fukuda test, we obtained average values below 30 degrees for both genders (this being the lower limit that signals the manifestation of vestibular disorders on the side where the body rotation is manifested). However, several individual cases exceeding this threshold should also be reported: 10 men (9.17%) and 6 women (12.5%). For men, we notice nine cases with a lack of body rotation (0 degrees), and for women, only one. Both genders have a higher incidence of body rotation on the right side; 70 men (64.22%) and 33 women (68.75%). Even though the average results do not indicate the presence of vestibular disorders, most students could not keep the body at the same starting point, being signaled forward movements when successively raising the knees up, with eyes closed, isolated cases moved forward a length of 2 m.

The average values related to better results belong to the groups of athletes, especially those who practice team sports games (TSG); they obtain results superior to the group of non-athletes (NA) in all tests, with evidence of statistically significant differences (*p* < 0.05) for five of the tests: the One-leg standing balance test, Stork test, Flamingo test, Walk and turn field sobriety test, and Fukuda test. The situation is similar for the differences between the individual sports (IS) and non-athletes (NA) groups, but here we find significant differences in only two cases: the Stork test and the Fukuda test. Even if the TSG group performs better in all equilibrium tests than the IS group, these differences are not statistically confirmed for the analyzed pairs, all being insignificant (*p* > 0.05).

The weaker average results of the NA group for the One-leg standing balance test are also generated by the low number of cases with higher values (over 20 s): only four cases (5.79% for the NA group), compared to seven cases (15.55% for the IS group), and eight cases (18.60% for the SG group). However, the situation is more balanced in the case of the Functional reach test, where the superiority of the TSG and IS groups is not statistically confirmed (*p* > 0.05), but the TSG group still has the highest values (greater than 50 cm): 11 cases (25.58% within this group), compared to 9 cases of the NA group (representing 13.04%), and 6 cases of the IS group (with 13.33%). For the Stork test, the significant differences (*p* < 0.05) reported between the NA group and the two groups of athletes are explained by the presence of higher individual scores for these two groups but also by the large number of poor results (less than 2 s of holding) recorded at the level of the NA group: 24 cases (i.e., 34.78%), compared to 4 cases for TSG (i.e., only 9.30%), and 9 cases for IS (i.e., 20%).

For the Flamingo test, we noticed a lower number of cases that have more than 10 falls for the TSG group (one participant, i.e., 2.32%), unlike the IS group (five subjects, i.e., 11.11%) and the NA group (eight cases, i.e., 11.59%). In this test, we also report higher individual values (which manage to maintain balance in a single attempt) of the TSG group (eight cases, i.e., 18.60%) compared to those in the IS group (six cases, i.e., 13.33%), but the difference between the average values of the two groups is not statistically significant (*p* > 0.05). The bass test is the second test where the superiority of TSG and IS groups over NA is not statistically confirmed, even if very good scores (over 90 points) are reported for athletes groups: 13 cases, 30.23% for SG, 7 cases, 15.55% for IS, and 8 cases, only 11.59%, for the NA group.

A higher number of cases (12 subjects) in the NA group make mistakes during the assessment of balance during walking (Walk and turn field sobriety test) compared to the groups of athletes (1 case for TSG and 3 cases for IS). At the level of the TSG group, 42 students follow the route without error (97.67%); for the IS group, this aspect is also confirmed for 42 participants (93.33%), while for the NA group, the number of those without errors is 57 cases (only 82.67%), which confirms the superior postural stability of athletes during dynamic actions. The assessment of body stability by the Fukuda test indicates significantly better values for the two groups of athletes (*p* < 0.05) compared to the NA group. At the level of the NA group, 10 cases with values higher than 30 degrees are identified (14.49%), for the TSG group, there are only 3 cases (6.97%), and for the IS group, there are also 3 subjects (6.66%). The lack of rotation around the body axis is reported for 10 subjects in the whole group, of which 3 are in the NA group (4.34%), 4 in the TSG group (9.30%), and 3 in the IS group (6.66%).

The comparative average values (by genres) of the balance evaluation for the NA group, are represented graphically in Figure 1, being in accordance with those analyzed in the comparison by gender. It is observed that the group of women has slightly superior performance in almost all tests: they get higher times in the One-leg standing test and Stork test, higher values in the Functional reach test, fewer falls in the Flamingo test, more points in the Bass test, and commit fewer errors in the Walk and turn field sobriety test. The only test where men in this category perform better is the Fukuda test, with an average value that indicates lower rotation around the body axis.

At the level of the TSG group, we identify changes in the distribution of performance between genders, but these results must be analyzed with reservation; due to the small group of women (only seven cases), according to Figure 2, men have a significantly higher score for the One-leg standing test and a slightly better value for the Stork test, which can also be explained by the greater force required to maintain positions in these tests. They also have superiority over Fukuda test values. The women maintain their superiority in the Flamingo test (where they perform a smaller number of falls), the Bass test, the Functional reach test, and the Walk and turn field sobriety test (where they walk the route without error).

Figure 3 presents the comparison between genders for the category of those included in the IS group. In this case, the men’s team maintains its superiority only for the One-leg standing test. The women confirm their superiority for the Flamingo test, the Bass test, the Functional reach test, and the Walk and turn field sobriety test, but in this case, they also slightly outperform the men in the Stork test and the Fukuda test. The comparison of these pairs should also be viewed with caution, without generalizing the conclusions, due to the small number of women in the IS group.

## 4. Discussion

We compared our results to those provided by similar studies, differentiating and grouping those that investigate the values of balance according to gender and those that compare different sports and groups that are not involved in practicing physical activities.

### 4.1. Comparison of Results with Similar Studies on Balance Values According to the Gender

The dynamics of the values of balance by age and gender indicate for the stage 20–49 years a consistency of results, followed by a reduction of values after 50 years, with men having slightly better results than women [62]. However, our research indicates superior male performance for only two tests (the One-leg standing balance test and Fukuda test, without reporting significant differences *p* < 0.05).

Decreasing sedentary behavior for young adults (x = 21 years) optimizes the values of static balance, a fact confirmed for those involved in moderate to vigorous PA, and for women, there are better values than for men, according to [63]. For the groups we studied, these aspects related to the superiority of women’s body stability are confirmed, regardless of the type of sports activity in which they are involved, especially for the Bass test, Flamingo test, Functional reach test, and Walk and turn field sobriety test (where the subgroups of women athletes follow the route without mistakes).

Female soldiers (Air Assault), with an average age of 26.4 years, show higher static balance values than men, but no differences are reported between genders for dynamic balance performances [64]. Dynamic stability testing during a single-leg jump landing did not differentiate between the results of the dominant leg vs. non-dominant leg, but gender differences are found, with women exhibiting higher dynamic postural stability scores in the vertical direction [65]. The postural balance of girls who practice alpine skiing is superior to boys of the same age (13 years) in the mid-lateral plane (ML), and subjecting both genders to the same training can reduce the differences between balance values, but not conditions of muscle fatigue [66]. Fatigue induced by high-intensity exercise applied to Irish university students creates problems of dynamic stability, but women outperform men before and after the exercise, being less affected by fatigue [67]. We cannot comment on the values of balance with fatigue background, as all the tests of our group were performed on the background of rest of the nervous and muscular systems.

For healthy young people in Romania (18–25 years), there are no differences in postural balance according to gender, but the group of those who are involved in moderate physical activity has higher values of balance compared to those who have a low level of physical activism [68]. For our group, we identified the superior performance of women for most tests, even if they are only significant for the Flamingo test and Bass test (*p* < 0.05). For the NA group (characterized by moderate effort), women also have superior results, with one exception: the Fukuda test.

A comparison of static balance results (single-leg stance position) for young athletes in Slovenia identifies higher values of dancers compared to other sports, and girls perform significantly better than boys [69]. The evaluation of Danish athletes (14–24 years old) on the One-leg standing test identified superior values of static balance for basketball players (positive associations with seniority in sports), but did not indicate significant differences between genders or age stages [70]. Sports activity improves the values of static balance (one leg standing with eyes closed and open) for paralympic goalball players. Higher values of balance are highlighted for those with more than 10 years of experience and for those who have reached the higher stages of competition, but no significant differences are reported between women and men [71]. Gender differences in maintaining static balance (one leg standing) are reported among NCAA Division III soccer players, with women outperforming men [72]. Our results still indicate a superiority of men for the One-leg standing test, but this is statistically insignificant (*p* > 0.05). Only at the level of the NA group are there slightly better values of women reported, but still insignificant; men have higher values for TSG and IS groups. When evaluating the static balance of the university students in our group, the girls have a slight superiority for the Stork test, but it is not statistically confirmed (*p* > 0.05), which is also maintained at the level of the NA and IS groups. However, men in the TSG group have higher values than women in the same category.

Female university students in physical education and sports (Poland) have higher values than men for the Flamingo test. The dynamics of 10-year results (2004–2010) showed a significant decrease in the results for men and an insignificant decrease for girls [73]. Our study confirms the statistical superiority of women for the Flamingo test (*p* < 0.05), which is also highlighted by the comparison between genders for each type of physical activity (NA, TSG, IS), with women needing fewer attempts to complete the test in all cases. 

Research on team sports players (basketball, volleyball, and korfball team) showed higher values of DSI (dynamic stability index) scores for men in most cases, so they have significantly lower dynamic stability than women [74]. The group of men in our research has higher values in the dynamic tests and better vestibular stability only for the Fukuda test, but statistically insignificant (*p* > 0.05) aspects were confirmed for those belonging to the TSG and NA groups. For the Walk and turn field sobriety test, women make fewer mistakes but still without significant differences.

Gender comparative studies have shown that female performance gymnasts have superior postural stability compared to male gymnasts via posturographic analysis, according to [75]. The better balance of girls at the Academy of Physical Education (Poland) is manifested both in quiet conditions and in the presence of acoustic noise (60–80–100 dB), with eyes closed or open, having lower CoP displacements than those of men [76]. The testing of the participants in our group was also performed in quiet conditions, but we cannot make a comparison with the results of the gymnasts due to the absence of this specialization from the study.

### 4.2. Comparison of Results with Similar Studies Related to Balance Values Depending on the Specifics of the Sports Activity

Although the demand for balance varies from one branch of sport to another, no statistically significant differences are found between team sport game practitioners and those involved in individual sports [77]. For our group, the TSG group has higher average values for all tests, but these are not statistically confirmed (*p* > 0.05).

For Tunisian teenager football players (U-17), the application of an 8-week workout (based on plyometrics and short sprints) resulted in significant performance improvements related to lower limb strength, agility, and static balance assessed by the Stork test [78]. Women who play football have a higher postural balance than those who are sedentary, better supporting the asymmetrical distribution of body mass and integrating vestibular information faster [79]. A comparative study of the static balance on one leg between the practitioners of several sports highlighted the superior stability of soccer players (who have a lower balance in the antero-posterior and horizontal plane), compared to swimmers, basketball players, and non-athletes [80]. Our study confirms slightly higher scores on the One-leg standing test for the TSG group, compared to IS, but without statistical significance (*p* > 0.05). Instead, significant differences are reported in this test for the TSG group compared to the NA group (*p* < 0.05).

Using an occlusal device for professional ballet dancers (Italy/Milan) for 6 months improved neuromuscular coordination and static balance performance in the Flamingo test [81]. Additional programs based on balance exercises have a positive effect on the postural stability of Czech Republic university students specializing in gymnastic sports, assessed by the Flamingo test on the dominant and non-dominant foot [82]. For students specializing in Physical Education and Sports (Turkey), the efficiency of artistic gymnastics planning is highlighted in terms of the values of static balance obtained in the Flamingo test [83]. The assessment of the balance of the athletes by the Flamingo test highlighted the statistically significant superiority of the tennis players for both legs compared to the soccer, basketball, and volleyball teams, between which there are no significant differences, according to [84]. In our group, the small number of tennis players (three cases) does not allow us to make a conclusive comparison, and we are aware that the presence of gymnasts would have radically improved the average values of IS. We notice a superior and significant stability in the Flamingo test for the TSG group compared to the NA group (*p* < 0.05), but we did not obtain significant differences between the TSG and NA groups (*p* > 0.05).

Investigations on university sports students in Japan have confirmed their superiority in postural stability during head rotation and better visual acuity compared to the non-sports group [85]. Regardless of the sport practiced, the values of balance are superior to those not involved in sports activities. Strong correlations between body stability and performance values are identified for shooters [86]. A study made on 6–47-year-old athletes showed that regardless of the sporting activity practiced (shooting, football, boxing, cross-country skiing, gymnastics, running, team games played with hands, wrestling, tennis, alpine skiing, rowing, and speed skating) they have superior postural stability compared to non-athletes in the bipedal position with their eyes open and closed (lower swing of the CoP) [87]. Our results confirm these ideas because the NA group has the poorest results compared to the TSG and IS groups.

The types of sports influence the value of performances in balance tests. A study on Iranian sports students showed that the best results in static balance are achieved by gymnastics, then martial arts, soccer, wrestling, basketball, and weightlifting. On top for the dynamic balance are wrestling, followed by soccer, gymnastics, weightlifting, and basketball [88]. Elite athletes who practice rowing have higher static stability (bipedal and monopodial) than field sports athletes, as a result of a superior adaptation of vestibular and proprioceptive functions to training requirements [89]. The comparison between a group of young gymnasts and a group of non-gymnast girls from Slovenia highlighted the superior values of postural control for gymnasts, who have a reduced balancing and velocity of the CoP in the antero-posterior direction [90]. Experienced Polish gymnasts (seniors) are characterized by superior values of balance for standing positions and handstands, compared to younger gymnasts (juniors), so the long-term practice of this sport will ensure a higher level of static balance [91]. The postural stability analysis at the level of women athletes identified superior values of technical sports practitioners (first, gymnastics, then ski jumping and diving) by comparison with basketball athletes and leisure/recreational athletes [92]. The absence of these types of sports in our group does not allow us to comment on the data provided by the research mentioned.

Taekwondo practitioners achieve significantly better values of static balance (lower CoP balancing) through one leg standing on a strength plate compared to groups of soccer and handball athletes. The explanation would be the adaptation to high blows during training, according to [93]. Taekwondo training (applied weekly, 60 min, for 16 weeks) has beneficial and significant effects on the fitness level and static balance of pubescent children in South Korea, assessed by the Stork test [94]. Judo practitioners have values of balance by evaluating with closed eyes superior even to dancers, as a result of adapting their proprioceptive mechanisms. Both categories of athletes have superior results in the evaluation with open eyes of groups of non-athletes, confirming the favorable effect of sport on postural stability [95]. Practicing regular dancing among Polish university students (seniority of at least 7 years, with 3 workouts per week/6–7 h cumulated) provides superior balance to those who are not engaged in physical activity by testing with eyes closed and open, in bipedal and unipedal support variants [96]. The comparison between dancers with seniority of over 7 years and non-dancing women (18–23 years old) identifies significant differences in static balance for dancers and an increased incidence of loss of balance for the other group, confirming the beneficial role of therapy through dance, as a way to prevent falls and injuries [97]. The comparison of the values of the balance between dancers and non-dancers showed better performances of the dancers for maintaining the first ballet position (with heels touching and feet externally rotated to 140 degrees), according to [98]. There are also studies that do not confirm the existence of higher values of static and dynamic balance for those who practice Latin American dances, compared to groups that are not involved in this type of physical activity (Poland), according to [99]. We notice the above-average results, in most tests, of the four athletes who practice dance and of four of the six martial arts practitioners.

## 5. Conclusions

Sports specialization generates several situations of significant differences in the manifestation of balance compared to the gender variable; the performances obtained indicate the level of demand for static and dynamic stability, according to the sport practiced. The fact that the NA group has the lowest average values in all tests confirms the beneficial effect of sports activities on body balance.

Gender differences support a superiority of women for five of the tests applied, which is in agreement with many of the analyzed studies, but significant differences and confirmation of the H1 hypothesis are obtained only for the Flamingo test and Bass test (*p* < 0.05). Men perform better at the vestibular stability/Fukuda test and One leg standing test, but these differences are statistically insignificant (*p* > 0.05).

Even if the TSG batch has slightly better values than the IS batch for the whole set of tests, they are not statistically significant (*p* > 0.05), so the H2 hypothesis is not confirmed. The differences between these two groups must be viewed with caution, as the study lacks sports specializations (such as gymnastics and skating), which would certainly have improved the average values of the IS group.

We noticed significant differences in favor of the TSG group compared to the NA group; for most tests, the H3 hypothesis was confirmed (*p* < 0.05), except for the Functional reach test and Bass test (*p* > 0.05). The comparison between the IS and NA groups generates only two pairs where the H4 hypothesis is confirmed: the Stork test and Fukuda test, with (*p* < 0.05); in the other tests, insignificant differences were reported (*p* > 0.05).

The research results cannot be generalized due to the poor numerical and percentage representation of certain sports (for example, there is an imbalance between soccer players and other team sports), as well as the low number or lack of athletes in other sports types and trials less popular and less practiced (hockey, weightlifting, gymnastics, shooting, skiing, skating). This makes the comparison between the different branches of sport irrelevant in terms of balance performance. The impossibility of testing the batch using baropodometric platforms, motion sensors, or pressure insoles at the time of the study is another neuralgic point, these data being much more accurate in analyzing and evaluating CoP variations in different directions and planes of motion [100,101,102].

Future research directions are focused on: the role of anthropometric variables in the manifestation of body balance; evaluating the performance of university students on other components of fitness, for example, in tests of strength and agility, which also involve the manifestation of good balance (the study is already underway); investigating the variations of static and dynamic postural stability depending on the seniority in sports activity or affecting the performance of the balance depending on the level of fatigue generated by the different stages of the intensity of effort.

## Figures and Tables

**Figure 1 ijerph-19-07672-f001:**
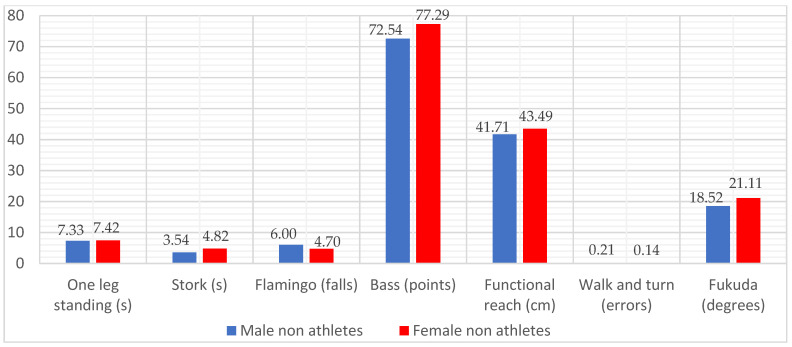
Distribution of average values in the balance tests for non-athletes (NA), differentiated by gender (male/N = 42, female N = 27).

**Figure 2 ijerph-19-07672-f002:**
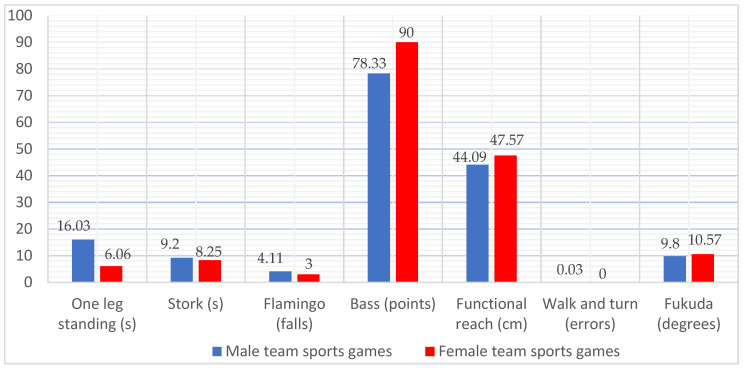
Distribution of average values in the balance tests for the group of team sports game practitioners (TSG), differentiated by gender (male = 36, female = 7).

**Figure 3 ijerph-19-07672-f003:**
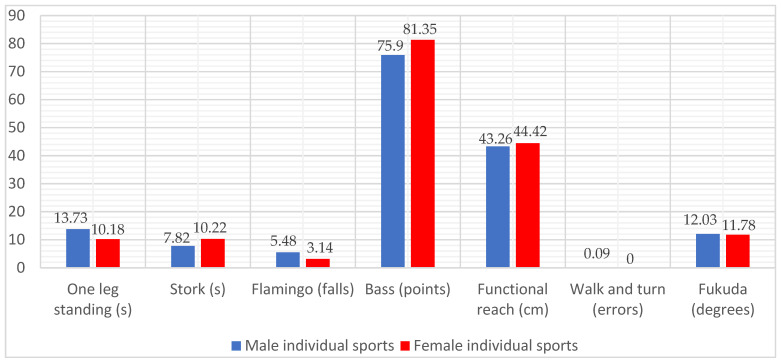
Distribution of average values in the balance tests for individual sports (IS) practitioners, differentiated by gender (male = 31, female = 14).

**Table 1 ijerph-19-07672-t001:** Distribution of participants and related percentages according to gender and specific sports activity.

Gender	Participants	Non-Athletes	Team Sports Games	Individual Sports
**Male**	109 (69.42%)	42 (26.75%)	36 (22.93%)	31 (19.74%)
**Female**	48 (30.58%)	27 (17.20%)	7 (4.46%)	14 (8.92%)
**Total lot**	157 (100%)	69 (43.95%)	43 (27.39%)	45 (28.66%)

**Table 2 ijerph-19-07672-t002:** The results of the Multivariate Tests ^a^ (MANOVA).

Effect	λ	F	Hypothesis df	Error df	Sig.	Ƞ^2^_p_	Observed Power
Gender	0.892	2.508 ^b^	7.000	145.000	0.018	0.108	0.867
Sport activity	0.804	2.386 ^b^	14.000	290.000	0.004	0.103	0.979
Gender * Sport activity	0.957	0.465 ^b^	14.000	290.000	0.950	0.022	0.288

^a^ Design: Gender + Sports activity + Gender * Sports activity; ^b^ Exact statistic; λ—Wilk’s lambda; F—Fisher test; df—degrees of freedom; Sig.—level of probability; Ƞ^2^_p_—partial eta squared.

**Table 3 ijerph-19-07672-t003:** Univariate test results (ANOVA)—The influence of the gender variable on the results of balance tests.

Dependent Variable	Sum of Squares	Mean Square	F (1, 155)	Sig.	Partial Eta Squared	Noncent Parameter	Observed Power
One-leg standing balance test	531.578	531.578	3.007	0.085	0.019	3.007	0.407
Functional reach test	67.071	67.071	1.574	0.211	0.010	1.574	0.239
Stork test	2.356	2.356	0.028	0.868	0.000	0.028	0.053
Flamingo test	50.365	50.365	4.803	0.030	0.030	4.803	0.586
Bass test	806.835	806.835	4.499	0.036	0.028	4.499	0.559
Walk and turn field sobriety test	0.043	0.043	0.390	0.533	0.003	0.390	0.095
Fukuda test	311.226	311.226	1.606	0.207	0.010	1.606	0.242

**Table 4 ijerph-19-07672-t004:** Univariate test results (ANOVA)—The influence of the sports activity variable on the performance of balance tests.

Dependent Variable	Sum of Squares	Mean Square	F (2, 154)	Sig.	Partial Eta Squared	Noncent Parameter	Observed Power
One-leg standing balance test	1522.142	761.071	4.438	0.013	0.054	8.875	0.756
Functional reach test	138.250	69.125	1.630	0.199	0.021	3.259	0.340
Stork test	882.374	441.187	5.500	0.005	0.067	11.000	0.845
Flamingo test	65.282	32.641	3.122	0.047	0.039	6.243	0.593
Bass test	928.811	464.406	2.584	0.079	0.032	5.168	0.509
Walk and turn field sobriety test	0.832	0.416	3.923	0.022	0.048	7.845	0.700
Fukuda test	2930.903	1465.452	8.234	0.000	0.097	16.467	0.958

**Table 5 ijerph-19-07672-t005:** Analysis of the significance of the differences between the average values in the balance tests according to gender (male = 109, female = 48).

Test/Dependent Variables	Group	Minimum	Maximum	Mean	Std. Deviation	Std. Error	a–b	Sig. ^b^
One-leg standing balance test	a. Male	1.85	89.18	12.0264	15.47649	1.274	3.994	0.085
	b. Female	2.39	26.62	8.0325	5.71390	1.919		
Functional reach test	a. Male	21.00	59.00	42.9417	7.34700	0.625	−1.419	0.211
	b. Female	37.00	53.00	44.3604	4.05787	0.942		
Stork test	a. Male	1.00	82.84	6.6308	10.47051	0.885	−0.266	0.868
	b. Female	1.22	24.79	6.8967	5.44521	1.334		
Flamingo test	a. Male	1.00	14.00	5.2294	3.46579	0.310	1.229 *	0.030
	b. Female	1.00	12.00	4.0000	2.64173	0.467		
Bass test	a. Male	34.00	100.00	75.4128	14.10884	1.283	−4.920 *	0.036
	b. Female	46.00	97.00	80.3333	11.57645	1.933		
Walk and turn field sobriety test	a. Male	.00	2.00	0.1193	0.35289	0.032	0.036	0.533
	b. Female	.00	1.00	0.0833	0.27931	0.048		
Fukuda test	a. Male	.00	65.00	13.7982	13.81276	1.333	−3.056	0.207
	b. Female	.00	60.00	16.8542	14.16016	2.009		

* The mean difference is significant at the 0.05 level. ^b^ Adjustment for multiple comparisons: Bonferroni.

**Table 6 ijerph-19-07672-t006:** Analysis of the significance of the differences between the average values by categories of sports activities in the balance tests (non-athletes/NA = 69, team sports games/TSG = 43, individual sports/IS = 45).

Test	Group	Minimum	Maximum	Mean	Std. Deviation	Std. Error	a–b	Sig. ^b^	a–c	Sig. ^b^	b–c	Sig. ^b^
One-leg standing balance test	a. NA	1.85	34.57	7.370	6.758	1.577	−7.039 *	0.019	−5.257	0.113	1.782	1.000
	b. TSG	2.55	57.50	14.409	16.120	1.997						
	c. IS	1.91	89.18	12.627	16.781	1.952						
Functional reach test	a. NA	22.00	54.00	42.410	5.879	0.784	−2.252	0.231	−1.214	0.997	1.038	1.000
	b. TSG	21.00	59.00	44.662	7.656	0.993						
	c. IS	29.00	57.00	43.624	6.250	0.971						
Stork test	a. NA	1.00	26.00	4.042	4.378	1.078	−5.006 *	0.014	−4.531 *	0.027	0.474	1.000
	b. TSG	1.00	41.72	9.047	8.469	1.366						
	c. IS	1.12	82.84	8.573	13.515	1.335						
Flamingo test	a. NA	1.00	14.00	5.492	3.432	0.389	1.563 *	0.042	0.737	0.708	−0.825	0.700
	b. TSG	1.00	11.00	3.930	2.501	0.493						
	c. IS	1.00	13.00	4.755	3.523	0.482						
Bass test	a. NA	46.00	100.00	74.405	12.234	1.614	−5.827	0.080	−3.194	0.647	2.633	1.000
	b. TSG	46.00	100.00	80.232	14.430	2.044						
	c. IS	34.00	95.00	77.600	14.102	1.998						
Walk and turn field sobriety test	a. NA	0.00	2.00	0.188	0.429	0.039	0.165 *	0.030	0.122	0.159	−0.043	1.000
	b. TSG	0.00	1.00	0.023	0.152	0.050						
	c. IS	0.00	1.00	0.066	0.252	0.049						
Fukuda test	a. NA	0.00	65.00	19.536	15.221	1.606	9.606 *	0.001	7.581 *	0.011	−2.025	1.000
	b. TSG	0.00	50.00	9.930	11.187	2.035						
	c. IS	0.00	60.00	11.955	12.058	1.989						

* The mean difference is significant at the 0.05 level. ^b^ Adjustment for multiple comparisons: Bonferroni.

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
