# Peer review of "The Influence of Gender and the Specificity of Sports Activities on the Performance of Body Balance for Students of the Faculty of Physical Education and Sports"

_ijerph, 2022, doi:10.3390/ijerph19137672_

Round 1
Reviewer 1 Report
I have just read this paper with high intrigue. It is fascinating to know how sport activity practice and sex can affect body balance, what is so important in the majority of sports. For that reason, I commend the authors for their effort. Nevertheless, I do have few minor concerns for the authors to review. I hope the authors will find these suggestions helpful and they could improve the great manuscript done.
General
· I am not sure the word gender should be used in this study, it seems that it refers to the word sex.
Abstract
· P of p-value must appear always in small letter.
Introduction
· From line 54 to 62. The statement is not clear; at the beginning you write that anthropometry is not related to balance. Later you say they are related and after that you write again they are not related. So what is the current statement in relation between balance and body characteristics?
· Line 71. What is CoP?
· From line 73 to 75. The statement "Adolescent sports girls have a dynamic postural control superior to non-sports girls (medial, posteromedial, and posterior directions), after performing an effort of 20 minutes" is breaking muscle or body fatigue cohesion in the paragraph. I suggest putting it in other place of the paragraph or relate the statement more.
· Line 87. Rewrite the sentence adding "Furthermore, the existence of groin pain..."
Materials and methods
· From line 243 to 247. I advise to explain deeper the functioning of these balance tests in order to improve reader understanding.
· Line 269. Correct word "tets".
Results
· From line 336 to 368. I suggest reducing these paragraphs since it does not show significant results for this items. Perhaps you are explaining too much this case.
· From line 382 to 417. I suggest reducing these paragraphs since it does not show significant results for this items. Perhaps you are explaining too much this case.
Discussion
· Part 4.1. I suggest connecting the previous articles clearer with the results presented in this study. By commenting on all previous results in a list form, one can lose the way they relate present study and previous ones. Perhaps, writing them all at once without recapitulating in comparison with the results of this study can diminish the reader’s understanding.
Reviewer 2 Report
Dear authors, I reviewed your paper -The influence of gender and the specificity of sports activities on the performance of body balance for students of the Faculty of Physical Education and Sports - and I found the topic interesting.
There are some issues that you must solve.
Your target age group is 20±2 years, but in Introduction you presented studies from other age groups. I recommend to exclude those references (lines 63-69, 107-125, 136-137, 175-181, 185-193).
In table 1 there are small errors: first line the sum is 69.42 instead 69.43, same issue in line 2. Check again all calculation by line and column.
You used abbreviation in different ways for second (sec line 328, sec. lines 336-338).
- Acronyms/Abbreviations/Initialisms should be defined the first time they appear in each of three sections: the abstract; the main text; the first figure or table. When defined for the first time, the acronym/abbreviation/initialism should be added in parentheses after the written-out form.
- SI Units (International System of Units) should be used. All information from Instructions for Authors.
Same issue for figure in text (Fig. 1, 2,3, lines 422, 435, 447). - There are mentioned authors in text - line 465.
In Discussion exclude studies that mentioned other group ages (lines 465-466, 485-490, 500-502, 509-512517-519, 546-548, 555-557, 562-566, 572-575).
Exclude subtitle - Limits of the study and future directions of investigation from Conclusion.
At Informed Consent Statement you must mention that subjects agreed to participate in the study.
At References the space between the lines must be smaller.
